# Connections between Different Sports and Ergogenic Aids—Focusing on Salivary Cortisol and Amylase

**DOI:** 10.3390/medicina57080753

**Published:** 2021-07-26

**Authors:** Cezar Honceriu, Alexandrina-Stefania Curpan, Alin Ciobica, Andrei Ciobica, Constantin Trus, Daniel Timofte

**Affiliations:** 1Faculty of Physical Education and Sports, Alexandru Ioan Cuza University, B dul Carol I, No 11, 700506 Iasi, Romania; chonceri@yahoo.fr; 2Department of Biology, Faculty of Biology, Alexandru Ioan Cuza University, B dul Carol I, No 11, 700506 Iasi, Romania; andracurpan@yahoo.com; 3Center of Biomedical Research, Romanian Academy, B dul Carol I, No 8, 700505 Iasi, Romania; 4Academy of Romanian Scientists, Splaiul Independentei nr. 54, Sector 5, 050094 Bucuresti, Romania; 5Department of Physiology, “Grigore T. Popa” University of Medicine and Pharmacy, 16, Universitatii Street, 700115 Iasi, Romania; thereau_86@yahoo.com (A.C.); dantimofte@yahoo.com (D.T.); 6Department of Morphological and Functional Sciences, Faculty of Medicine, Dunarea de Jos University, 800008 Galati, Romania

**Keywords:** sports, ergogenic aids, salivary cortisol, salivary amylase

## Abstract

Athletes are exposed to a tremendous amount of stress, both physically and mentally, when performing high intensity sports with frequent practices, pushing numerous athletes into choose to use ergogenic aids such as caffeine or β-alanine to significantly improve their performance and ease the stress and pressure that is put onto the body. The beneficial or even detrimental effects of these so-called ergogenic aids can be appreciated through the use of numerous diagnostic tools that can analyze various body fluids. In the recent years, saliva samples are gaining more ground in the field of diagnostic as it is a non-invasive procedure, contains a tremendous amount of analytes that are subject to pathophysiological changes caused by diseases, exercises, fatigue as well as nutrition and hydration. Thus, we describe here the current progress regarding potential novel biomarkers for stress and physical activity, salivary α-amylase and salivary cortisol, as well as their use and measurement in combination with different already-known or new ergogenic aids.

## 1. Introduction

It is a well-known fact that exercising even to the mildest intensities brings a plethora of health benefits to the individual, both mentally and physically. However, it is also accepted that exercising triggers the body’s stress responses mechanisms. These types of responses can be quantified by the use of biomarkers, more precisely the measuring of α-amylase and cortisol from saliva samples as a surrogate measure of psychophysiological stress. In recent years, saliva samples are gaining more ground in the field of diagnostic as it is a non-invasive procedure, contains a tremendous amount of analytes that can be subjected to pathophysiological changes caused by diseases, exercises, fatigue as well as nutrition and hydration. Also, the chemical composition of saliva is rather simple, which makes the analysis less tedious and less time-consuming, but it is precise enough to detect hormone differences based on diet, age, sex and activity (*p* < 0.001) [1].

Numerous studies have demonstrated the use of salivary biomarkers as diagnostic candidates for measuring physiological and psychological stress in response to various exercises. A significant relationship was observed between salivary and serum levels [2], reinforcing the idea that saliva is just as a valuable fluid as blood.

The human body attempts to maintain a physiological stability when facing general stress signals through the activation of adaptive mechanisms formed by the hypothalamic-pituitary-adrenal (HPA) axis and sympathetic adrenomedullary system (SAM) which influence parameters such as immunoglobulin levels, salivary cortisol and amylase in humans. Substantial evidence indicates that the chronic elevation of cortisol levels and the dysfunction of the feedback system within the HPA axis play a prominent role in stress responses [3,4]

Stress is reported to have minor to major effects on, and in a sports environment, levels of stress have been correlated with the recovery time, burnout due to overtraining, the time to repair muscle damage, and the susceptibility to infection [4].

Salivary cortisol, salivary immunoglobulin A and the associate salivary flow rate are affected by the daily rhythms. It is well understood that the first sight of daylight after waking initiates the circadian response although there is surprisingly little research available on daily rhythm variation of the normal salivary flow rate over a 24 h period [4].

The stress placed on the body through exercise can be measured by the abovementioned biomarkers. Two key systems in the neuroendocrine response to stress are the hypothalamic-pituitary-adrenal axis and the autonomic nervous system (ANS) that stimulate responses to assist the body in coping with the applied stressor [5].

The main stress systems that are activated by exercising are the sympathoadrenal medullary system (SAM) and the hypothalamic pituitary adrenal axis (HPA). An increased sympathetic nervous activity can be appreciated by using a non-invasive biomarker capable of illustrating physiological stress–salivary α-amylase which has been shown to increase in response to acute stress [2]. However, since exercising is a source of stress for the body, both salivary cortisol and salivary amylase are capable to respond [6]. Modifications in the blood cortisol seems also to be relevant in this context [7].

Cortisol is a glucocorticoid released from the adrenal cortex after the activation of the HPA axis as a response to stress. Cortisol is capable of coping with stress and repair tissue damage through mechanisms such as protein and lipid catabolism stimulation, gluconeogenesis increase, and inflammation reduction. Importantly, when the HPA axis is chronically stimulated, the typical responsiveness of the axis may be suppressed, which may inhibit immune function. The responsiveness of the HPA axis makes cortisol a useful marker for assessing the acute (short-term) and chronic (long-term) response to stress in individuals and athletes [5].

Salivary cortisol is a good physiological biomarker for assessing stress level and hypothalamic-pituitary-adrenocortical axis function. The cortisol concentration changes in response to exercise and competition related stress. In fact, the free testosterone to cortisol ratio is used for assessing metabolism in relation to anabolic-catabolic balance. 

Psychophysiological stress, especially due to high-intensity exercise, leads to the activation of the ANS which in turn favors the secretion of α-amylase directly into the saliva making it a potential useful indicator of stress. Therefore, the salivary concentration is highly dependent on the ANS activity as well as the circadian rhythm since it is produced within the salivary glands whereas cortisol it is transported from plasma. However, when this particular biomarker is chosen as an indicator for ANS activity, the fact that it correlates poorly with other indicators (such as catecholamines) must be taken into consideration [5].

The circadian rhythm influences the levels of these parameters, as well as testosterone, in the way that cortisol and testosterone reach their peak at the beginning of the day, whereas amylase reaches it in the afternoon [2].

At high level sports events, where competition is at its finest with evenly matched opponents, even the smallest factors can make a significant difference in the outcome, either be it hydration, nutrition, sleep or mental stress level. Since not all athletes know the impact of dietary choices, between 40% and 100% of them opt for using supplements as a way of gaining an advantage of any kind [8]. Considering the fact that supplements are treated as a subcategory of food, which means they are not heavily regulated for safety and efficacy measures, their effect on performance and overall health might be actually detrimental, with some studies even reporting severe side effects from certain dietary supplements [8,9]. Great interest is currently placed on supplements with potential ergogenic effects, but there is not sufficient scientific evidence to back their performance enhancing abilities or their action on physiological biomarkers such as the abovementioned ones [10].

Caffeine as one of the most studied ergogenic substances, capable of improving performance and cognition even in sleep-deprived individuals [11]. β-Alanine may play a crucial role in reducing muscle mass and function, and also have a neuroprotective and antioxidant property which may result in enhanced exercise capacity [12]. Carbohydrates maintain blood glucose levels and may accelerate recovery after high intensity exercise [13]. Creatine may improve mood and the performance of tasks that place a heavy stress on the prefrontal cortex in sleep restricted individuals [14,15]. Some probiotic strains reduce the incidence of respiratory and gastrointestinal infections in athletes and therefore may improve exercise performance [16]. Sodium bicarbonate or sodium citrate may provide the body with extra endurance against fatigue through deleterious changes in acid-base balance [17,18].

For this reason, we have decided to analyze and put together a review regarding the effects of several claimed ergogenic substances on body, performance, endurance, and stress correlated with various factors such as sex, age, and the intensity of the physical activity through cortisol and amylase levels measurements.

## 2. Methods

In this review, a search in Medline and Hindawi databases was performed by using the keywords and all combinations of them (“sports”, “ergogenic aids”, “salivary cortisol” and “salivary amylase”). Articles were selected based on the keywords’ presence in the titles or abstracts. From the general articles on sports, ergogenic aids and stress system responses we have included in the search criteria specific different ergogenic substances presented in the current study such as caffeine, β-alanine, carbohydrates, sodium bicarbonate and probiotics. We have looked for studies including cortisol and amylase levels measurements where possible and also followed differences between age groups and biological sex. We included research articles, with a focus on human studies, and reviews published up to May 2021. We excluded studies measuring amylase and cortisol from blood samples as the purpose of our study was also to illustrate the use of salivary samples as a non-invasive method of analysis, but for the newer ergogenic aids (such as probiotics) we have included studies that only showed their effect on performance since we consider is of interest for current and future studies to illustrate what areas are still lacking and those were the only studies available at the moment we concluded our search.

## 3. Results

We have included 105 studies and reviews on the subject of this article in an attempt to highlight how different factors influence performance and overall health, their repartitions based on general aspects, sports & age/gender/caffeine adm./β-alanine adm./carbohydrates adm./sodium bicarbonate adm./probiotics adm. can be observed in Figure 1.

The summarized results on demographic groups as well as ergogenic substances administration can be visualized in Table 1, with more details that are to be discussed in the following paragraphs.

Hormones, such as glucocorticoids (especially cortisol due to its influence on human skeletal muscles) are altered by a series of factors as environment, nutrition, age, sex, overall health, genetics, and exercise. Exercise of any type has the ability to alter hormone secretion depending on the frequency, duration and intensity [19].

## 4. Demographic Groups and Sports

### 4.1. Age

Although exercising has a plethora of benefits on the individual of any age, it plays a crucial role in aging populations’ health. It is a well-known fact that exercising of various types of intensities can prolong life, improve cardio-metabolic health, muscle health, physical independence and all-cause mortality therefore exercise is a well-established preventive method against the detrimental effects of aging [69].

A study conducted by Azarbayjani et al., about the effects of exercise on salivary testosterone to cortisol ratio and alpha-amylase in young active males (healthy condition with three exercise sessions per week) has revealed decreased cortisol concentration in treadmill sessions and increased in elliptical and cycle ergometer but with no significant changes, whereas amylase activity has been deemed significantly declined after exercise on the elliptical instrument (*p* = 0.04) and on the treadmill (*p* = 0.006) at an intensity of 85% maximum heart rate. There was also a significant decrease of free testosterone to cortisol ratio post exercise (*p* = 0.01). Prior to tests, volunteers participated in a 5-min warm up composed of treadmill at two speeds but same gradient (6 km/h and 8 km/h with a 3% gradient) and they also drank 500 mL of water before the sampling in order to maintain normal hydration. However, no significant differences have been observed between cortisol and amylase, except that they were inversely related in the majority of cases. These results are similar to another study that observed higher cortisol level following high intensity exercises (*p* = 0.05) whereas the low intensity and control group showed no significant changes, but interesting when the working memory task was performed, only the low intensity group exhibited significantly improved cognitive performance. However, when the groups were split once again into low and high performers, significant improvements have been noted for the low performers of the both intensities groups however no correlation have been found between cortisol levels and working memory performance [70]. A study using repeated bouts of short-term and high-intensity cycling exercise on healthy teenager boys has also noted significantly increased salivary cortisol [71].

Interestingly, in a study conducted on middle aged man, the level of salivary cortisol has indeed increased post resistance exercise (assessed by using 1RM test in bench press, supported barbell row, squat and 45° leg press) in the untrained group (*p* < 0.05) but with no significant difference between the trained and untrained group of middle-aged individuals [20].

All the above results suggest that salivary cortisol levels are significantly modified when the individuals are exposed to a higher intensity of exercise than normal which supports the idea that the body is able to adapt to stress and reduce the response of the stress mechanisms.

### 4.2. Biological Sex

It is a commonly accepted fact that there are physiological and morphological differences based on the biological sex between males and females that can be easily monitored and observed in the body’s response specificity to ergogenic substances as well as different intensity and even different types of exercise according to the stress systems reaction.

The coordination of the mentioned stress systems pathways is a clear indicator of a functional stress response, while the activation of one without the other pinpoints towards a dysfunctional response. Interestingly, one study that measured salivary amylase and cortisol at rest and exercise both in female and male, have noticed a positive coordination for males and negative for females with the latter exhibiting a significant increase in amylase activity post exercise [1]. Nevertheless, such a result requires further studies in order to explain the negative trend for females as hormonal causes have been excluded in different previous studies.

Another study conducted on female and male athletes on the effects of 20 min high intensity interval training have observed a 1.5× higher amylase activity in males at rest, but similar cortisol levels, whereas post exercise there were no significant differences between the groups [21]. These observations are different from the ones noted by another group that used a 470 km journey (The Ecomotion/ProAdventure Race World) to assess stress responses, they have illustrated significant higher relative percentages of amylase and cortisol in women [22]. These results might be explained due to detrimental effects of a long-lasting competition on muscle, stress, and immune system. Similar results were obtained in a study performed on endurance trained athletes in a 5000 m race where basal amylase levels have been noted to be significantly higher in females [23]. These results reflect not only the influence of intensity on stress systems, but also the period individuals are engaged in the physical activity.

Interestingly, two studies performed during a Taekwondo competition and during the Korea National Shooters competition, have been able to highlight a possible connection between anxiety and salivary amylase concentrations [72], as well as cortisol concentration levels alongside its secretion rate [73] in both male and female, which may have a negative impact on performance. Therefore, amylase and cortisol have the potential to be easily used as an indicator in high level competitions in order to assess and improve athletes’ mental health and performance.

## 5. Ergogenic Supplements

### 5.1. Caffeine

Caffeine is a purine alkaloid which is naturally found in coffee, tea, guarana and cocoa and commonly used in a plethora of foods, beverages and even medications with its ergogenic properties being demonstrated by a number of studies. A dose of 3–6 mg/kg of body mass is recognized to improve performance [11,74]. Caffeine as well as its derivatives act as potent antagonists on adenosine receptor by inhibiting its negative effects on neurotransmission, arousal and pain perception and also determines increased levels of plasma catecholamines [11,75].

A study conducted on endurance athletes have reported a performance improvement when caffeine was administered 60 min prior or during exercising, but they also noted that the greatest chance of optimizing the ergogenic effect is a 7 days abstinence from caffeine before the use [33], while another study recommended a 14 days abstinence in the case of endurance performance in the scenario of a triathlon event [24]. Several studies conducted over the years have reported a significant performance increase following caffeine ingestion in a variety of sports, such as competitive intermittent-sprint [34,35], tennis performance [36], women’s rugby seven competition when administered under the form of energy drink [37], but did not impact the quality of technical actions’ [76] with another study stating that caffeinated coffee consumption failed to enhance time-trial performance in a 800 m-run in overnight-fasting runners [77]. One study have reported that caffeine intake combined with mental stress, which is the case for sport competitions, triggered a larger, prolonged response in men than in women in triathlon event performance [24], but another study reported enhanced endurance exercise performance in women with a magnitude similar to that in men [38]. Interestingly, when cognitive performance must be maintained under severe stress, caffeine consumption may provide a significant advantage [78].

Therefore, there is no surprise about caffeine being used in different sports also with effects other than performance improvement. Caffeine has been reported to also have a hypoalgesic effect by diminishing pain perception during exercise [39]. Caffeine directly affects the central nervous system through a decrease in the brain serotonin:dopamine ratio leading to delayed fatigue, improved motivation, alertness and vigilance. When it comes to cortisol levels, one study reported that microencapsulated caffeine led to higher post-triathlon cortisol levels [24], similar to a study on resistance exercise that also reported a decrease in testosterone:cortisol ratio [25], whereas caffeine administered under the form of caffeine gum have reported no changes in salivary cortisol after simulated half time by professional academy rugby union [26], but another study observed that it decreased fatigue during repeated, high-intensity sprint exercise in competitive cyclists associated with decreased salivary cortisol [79]. Interestingly, one study in particular noted caffeine administration determined higher adrenaline levels, higher cortisol levels after recovery leading to increased levels of IL-6 and IL-10 after treadmill exercise implying the involvement of the immune system [27].

When it comes to non-stressful conditions, one study has reported that acute coffee consumption (200 mL coffee containing 160 mg caffeine) activated salivary amylase, but not salivary cortisol suggesting the activation of the sympathetic nervous system and coffee’ potential as an antistress substance [28]. An increased sympathetic nerve activity may lead to a disruption in the post-exercise autonomic recovery following caffeine ingestion [80], and also caffeine delays parasympathetic recovery [81] and does not prevent the decline in performance in the morning nor improve performance in the afternoon in repeated spring exercise [82]. Consumption of green tea (caffeine 6 mh/kg) after a taekwondo training session determined higher salivary amylase activity when compared to the pre- and post-training levels [29]. Whereas caffeine consumed as a cereal bar during exhaustive cycling increased endurance and salivary cortisol but did not affect the salivary amylase increase post-exercise [30]. In the case of regular users of caffeine, lab administration of the substance did not observe any changes in salivary amylase, even after engaging in cognitive tasks [83].

A variety of studies have started to shift their attention on sleep importance and influence on performance especially in prolonged activities. Two studies have reported elevated salivary cortisol after caffeine administration in acute sleep deprived athletes and altered performance [31,32]. Interestingly, one study has reported that even the bitter taste of caffeine has an impact on the level of salivary amylase as they have observed that in the case of hypersensitive subjects the levels of amylase fragments were much higher [84].

### 5.2. β-Alanine

β-Alanine is an essential amino acid and a precursor in the carnosine synthesis pathway, a dipeptide highly concentrated in muscle and brain tissues especially during exercise; a diet supplemented with beta-alanine leads to an increased level of intramuscular carnosine, which in return functions as a buffer through its ability to locally accumulate H^+^ and modify the pH [85] resulting in better exercise performance and adaptation [41]. The concentration of muscle carnosine under normal conditions is highly dependent on age, sex, training status, eating habits, while β-alanine has been shown to increase muscle carnosine proportionally with the time of use [42,43].

β-Alanine has been recognized as a potent ergogenic aid with the most pronounced effects being observed in tasks that require ATP production via anaerobic glycolysis [86], therefore its benefits could potentially be noted in program with high repletion ranges or short rest periods between sets [87].

One study that analyzed the effects of β-alanine supplementation over a 3 week period during high-intensity anaerobic exercise in highly trained athletes have observed no significant improvements in fatigue rates; however they noted that as the time of administration continued, the efficacy became more apparent [44], whereas other studies observed improved biochemical parameters in regards to muscle fatigue [45] and increased exercise capacity [46] concomitantly with elimination of the decline of the executive function post-recovery [12]. A 4 week supplementation of β-alanine has successfully improved performance, marksmanship and target engagement speed, as well as muscle endurance [47], but with no impact on cognitive performance [48].

Muscle carnosine levels have been noted to increase at a 12 g/d^−1^ β-alanine supplementation [88] after an administration period of 30-days [89]. No influence on brain carnosine signal has been observed at 30 days or 28 days at a dose of 6.4 g/d^−1^ [89,90], but at a chronic dose of 22.5 mmol/kg under acute stressful conditions, β-alanine determined an increase of carnosine in cerebral cortex and hypothalamus, a decrease of a major metabolite of serotonin suggesting an anxiolytic-like effect of β-alanine [91].

The study conducted by Varanoske et al. focusing on β-alanine supplementation in a 24 h simulated military operation, highlighted an increased cortisol level and decreased testosterone:cortisol ratio at 24 h which indicates a catabolic state not influenced by β-alanine supplementation.

We were not able to identify any studies that measured salivary cortisol and salivary α-amylase upon supplementation with β-alanine and exercise/sports of different intensities, but β-alanine is a wide accepted ergogenic aid.

### 5.3. Carbohydrates

The diet composition has a big influence on the endocrine system responses in an active person. In the case of hypoglycemia, for example, the body determines an increase of the cortisol level as a countermeasure, therefore, it is important for athletes or individuals practicing sports as a recreational activity, to refill their carbs reserves before, during and after training [40,92].

Overtraining often exposes endurance athletes to the risk of upper respiratory tract infections as is producing higher plasma cortisol levels and greater immune disturbances and several authors have concluded that this risk can be attenuated by carbohydrates intake [49,50,51]. A study conducted by Costa et al. has illustrated that salivary cortisol has been observed to be similar in the self-selected diet group and carbohydrate group, but increased post-exercise in the self-selected diet group alongside a decreased blood glucose concentration. These results are similar to ones observed in the study of Soltani et al. that have reported a decrease of salivary cortisol levels as a result of increased dietary carbohydrate as part of a Dietary Guideline for Americans-based diet. As stress often met in competition is usually characterized by feelings of depression, tension, anger, anxiety, a high-carbohydrate diet has shown no effect of these, but it showed a higher salivary cortisol response in men [52,53,54]. Interestingly, when carbohydrate intake was supplemented with caffeine, the fat use was increase in a 20 km cycling time trial performance, but with no effect on performance [93].

### 5.4. Sodium Bicarbonate

High-protein diets have the tendency to produce an acid-base imbalance followed by an excessive production of endogenous acids in the body. Acidosis, characterized by increases in hydrogen ion production and decrease in blood pH levels, is a contributing factor to fatigue especially during high-intensity exercise. Acidosis can inhibit energy production leading to impaired power output and performance. For the purpose of decreasing acidosis and restoring the acid-base balance, several substances with buffering properties have been investigated, such as sodium bicarbonate which is capable of creating a more alkaline environment by increasing the levels of circulating bicarbonate [94,95]. The efficacy of sodium bicarbonate has been reported to be the most effective during high-intensity exercise that are mainly glycolytic, but high doses frequently leads to side effects such as nausea, diarrhea, stomach pain, and vomiting [96].

Sodium bicarbonate as an accepted extracellular buffer with the ability of enhancing performance, should, in theory, have more beneficial effects in combination with other ergogenic aids when it comes to high-intensity exercises. The review conducted by Naderi et al. analyzed the literature in search for the effects of co-ingestion of different ergogenic substances and has noted that sodium bicarbonate in combination with caffeine led to a longer total distance of rowing compared to sodium bicarbonate alone, or a possible additional benefit in 2000-m rowing performance when administered together with beta-alanine, while other studies have reported no-significant improvement of this co-ingestion, while sodium bicarbonate along with 6 days of creatine improved the second bout in free style swimming performance [55].

A study conducted on the effects of individualized sodium bicarbonate ingestion have highlighted a small, but significant performance effect and suggested that this effect was not due to meaningful differences in alkalinity [56]. Regarding the method of administration, one study has suggested a gelatin capsule and a dose of 0.3 g·kg^−1^ body mass of NaHCO_3_ 90 min before the exercise [97], while another study suggested the use of delayed-release capsules in other to reduce the gastro-intestinal side-effects and to further enhance its ergogenic effect [98].

The literature regarding the use of sodium bicarbonate as an ergogenic substance is still very limited with many articles illustrating contradictory results and with no studies analyzing salivary alpha amylase nor salivary cortisol. However, all the studies that we identified hint towards its possible potential and agree on its buffering abilities.

### 5.5. Probiotics

Probiotics are microorganisms administered for their protective abilities of the gut microbiota concomitantly with an antibiotic treatment. Prolonged exercise causes stress on the gastrointestinal tract which eventually leads to decreased performance and GI specific symptoms such as nausea, diarrhea, vomiting. The GI system influences HPA axis activation through feedback via the vagus nerve on the hypothalamus and hippocampus, with alteration in both systems being observed in irritable bowel syndrome and psychological disorders [57,99]. The literature supports the idea that some probiotics strains may regulate the immune response, improve the activity of macrophages, downregulate inflammatory cytokines and lower the risks for respiratory and gastrointestinal infections in athletes or active individuals. With these risks eliminated, probiotics might actually have an ergogenic effect, even if the underlying mechanisms are still unclear [16].

Studies have illustrated that probiotics may benefit athletes as lean body mass, normalized age-related declines in testosterone levels, reductions in cortisol levels have been noted. Some strains have been reported to even have an influence on recovery post-exercise [57,58]. As per literature search, multiple studies have confirmed the beneficial effects of probiotic supplementation in terms of intestinal permeability, immunity, microbiota, inflammation, GIs and upper-respiratory tract infections (URTIs) [59], with several being species or strain specific such as *Lactobaccilus*, *Faecalibacterium*–capable of easing inflammation, *Bifidobacterium*, *Bacteroides*, *Akkermansia*, *Faecalibacterium*–modulate immune system, *Saccharomyces*, *E. coli*–IBS, *Akkermansia*, *Eubacterium*–production of vitamin B12, with the latter being able to also improve insulin sensitivity and increase energy production [100].

Various studies have investigated the effects of different strains and species of probiotics in correlation with different types of sports and levels of activity. By using *Lb. fermentum* in elite male distance runners a decreased risk and severity of URTIs has been noted [60], while the same strain in male and female competitive cyclists led to decreased severity of GIT symptoms besides the effect on the respiratory tract [61]. The same strain co-administered with *Lb. acidophilus*, *Lb. rhamnosus*, *Lb. casei*, *Lb. plantarum*, *B. lactis*, *B. breve*, *B. bifidum* and *Streptococcus thermophilus* in 10 male runners increased tun time to fatigue and generated small to moderate improvement of gut permeability [62]. A study performed on quite an impressive number of participants (456 physically active males and females) by using *B. animalis* subsp. *lactis* BI-04 (BI-04), *Lb. acidophilus* NCFM and *B. animalis* subsp. *lactis* BI-04 (NCFM and BI-04) illustrated a decreased risk of URTIs by 27% [101]. In 30 male elite rugby players, *B. bifidum B. longum Lb. gasseri*, decreased the incidence of URTI/GIT [102]. The study conducted by Jäger, Purpura, et al. on 15 resistance-trained men treated with *B. breve* BR03 and *S. thermophilus* FP4 led to a positive effect on the reduced performance and range of motion followed by intense muscle damaging exercise [63], *Lb. acidophilus* (CUL60/CUL21), *B. bifidum* (CUL20), *B. animalis* subsp. *Lactis* (CUL34) decreased incidence and severity of GIT symptoms, both during training and a marathon race in 24 amateur athletes [103]. In the same number of participants, *Lb. rhamnosus* IMC 501 and *Lb. paracasei* IMC 502, decreased antioxidant levels followed by exercise [104]. While in 33 highly trained individuals, *B. bifidum* W23, *B.* W51, *Enterococcus faecium* W54, *Lb. acidophilus* W22, *Lb. brevis* W63, and *Lactococcus lactis* W58, decreased drops of tryptophan levels caused by intense exercise, decreased incidence of URTIs [105]. One study in particular has reported an improvement in moderate-intensity exercise following probiotics intake [64] while another has observed a decrease in Alzheimer’s disease progression [106], both conducted on mice. Other studies using *Lactobacillus plantarum* TWK10 reported significantly increased exercise performance in a dose-dependent manner [65], *L. plantarum PS128* as a potential ergogenic aid for better training management [66] or *Bifidobacterium longum* OLP-01 as an aid for performance-improving and health-promoting [67]. Interestingly, one study has reported the beneficial effects of probiotics in recovery with significantly increases at 24 and 72 h, as well as decreased soreness at 72 h post exercise [68].

## 6. Conclusions

The benefits of physical activity are without a doubt immense, but when a sport goes from being just a leisure and recreational activity to a highly competitive job, the body is exposed to enormous levels of stress best characterized and identified through different parameters. Athletes are subjected to close to impossible expectations and they often turn to approved and known ergogenic aids like the ones mentioned in the current study. In our article we have illustrated the current progresses regarding potential novel biomarkers for stress and physical activity, salivary α-amylase and salivary cortisol, as well as their use and measurement in combination with different already-known or new ergogenic aids. In some cases, literature has been limited, but that only highlights potential future research directions.

Despite the spectacular number of studies on this subject, all of them present several limitations: such as the duration of the study (usually just 4 weeks), the limited number the participants, no delimitation between before-trained and amateurs in the same study or the study itself has not been conducted under the umbrella of a randomized clinical trial, but the potential probiotics have shown is immense. Although the age is a determining factor for the body’ ability to perform certain tasks, the number of studies focusing on age related changes is rather limited. Also, it would be interesting to analyze why there is a negative trend for females as reason such as hormones or the menstrual cycle have been previously excluded.

All of these results reflect the influence of intensity on stress systems. In terms of salivary α-amylase and cortisol, the number of studies that measure both and correlate them is still reduced even if these systems are supposed to work together but studies on substances like caffeine have reported either increased cortisol, and an activated immune response either only amylase suggesting its potential as an antistress substance. However, caffeine also determines an increased sympathetic nerve activity which could lead to a disruption of the post-exercise autonomic recovery and does not prevent the decline in performance in the morning nor improve it in the afternoon.

At the same time, β-alanine has the best effects in tasks that require ATP production via anaerobic glycolysis therefore its benefits could potentially be noted in programs with short rest periods between sets. Under acute stress conditions, β-alanine is capable of increasing carnosine levels in the brain and decreasing a major metabolite of serotonin suggesting similar anxiolytic effects as caffeine. Often, cortisol levels are increased as a countermeasure to hypoglycemia therefore the diet composition plays a key role for body’ homeostasis and several authors have concluded that the risk of upper respiratory tract infections as a result of overtraining can be attenuated by carbohydrates intake, refilling the reserves before, during and after training. On the other hand, a high-protein diet can lead to acidosis which in turn inhibits energy production leading to impaired performance. Often, acid-base balance can be restored with the help of buffering substances such as sodium bicarbonate. However, at high doses it produces nausea, diarrhea, stomach pain and vomiting, but a study suggesting that by changing the form of the substances (e.g., gelatin capsules) can decrease the risks of adverse effects. This sort of symptoms is also specific to the gastrointestinal tract dysfunction observed after prolonged exercise, in irritable bowel syndrome or in psychological disorders. Even if probiotics are not the conventional class of probiotics, multiple studies have confirmed their beneficial effects on intestinal permeability, immunity, gut microbiota, inflammation and GI and upper-respiratory tract infections.

In conclusion, future studies should focus on the connection between α-amylase and cortisol in the context of exercise stress by using saliva samples since these are not an invasive method of analysis and the results obtained are similar to those obtained from blood samples. Before using ergogenic substances for increasing performance, the gastrointestinal tract health should be checked and maintained within normal values by means of a diet rich in carbohydrates or by using probiotics to improve overall immunity. Besides the performance improvements exhibited through ergogenic aids, they also seem to have an anxiolytic effect which in competition may be often more important than the performance aspect.

## Figures and Tables

**Figure 1 medicina-57-00753-f001:**
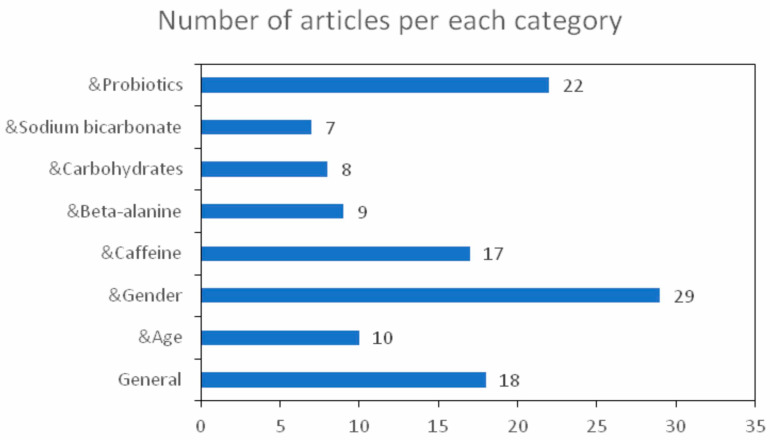
Number of articles used based on the category: sport- general aspects, sport & age, sport & gender, sport & caffeine administration, sport & beta-alanine administration, sport & carbohydrates consumption, sport & sodium bicarbonate administration and sport & probiotics administration.

**Table 1 medicina-57-00753-t001:** Summarized results of the papers included in this study.

Demographic Groups/Ergogenic Substance	Effect on Cortisol and Amylase Pre- and Post-Exercise	Effect on Performance
Age	↑ Cortisol, ↓ amylase in elliptical and cycle ergo meter in young males [2] ↓ cortisol, amylase in treadmill sessions in young males [2] ↑ salivary cortisol post resistance in middle-aged man [20] in superset strength training protocol	
Gender	↑ amylase activity post exercise in females in cycling [1] ↑ 1.5× amylase activity in males at rest, but similar cortisol levels in high intensity interval training [21] ↑ amylase and cortisol in females in the Ecomotion/ProAdventure Race World [22] -basal amylase levels higher in females in 5000 m race [23]	
Caffeine	↑ post-triathlon cortisol levels–microencalsulated caffeine [24] ↓ testosterone:cortisol ratio in resistance exercise [25] -caffeine gum–no changes in salivary cortisol after simulated half time by professional academy rugby union [26] ↓ salivary cortisol in repeated, high-intensity sprint exercise in competitive cyclists [26] ↑ adrenaline and cortisol levels after recovery leading to increased levels of IL-6 and IL-10 after treadmill exercise [27] -acute coffee consumption activates salivary amylase, but not salivary cortisol [28] -consumption of green tea after a taekwondo training session ↑ salivary amylase activity [29] -caffeine consumed as a cereal bar during exhaustive cycling ↑ endurance and salivary cortisol, but did not affect the salivary amylase increase post-exercise [30] ↑ salivary cortisol after caffeine administration in acute sleep deprived athletes and altered performance [31,32]	-performance improved in endurance athletes [24,33] -significant performance improvement in competitive intermittent-sprint [34,35], tennis performance [36], women’s rugby seven competition [37]. -enhanced endurance exercise performance in women [38] -hypoalgesic effect by diminishing pain [39] ↓ fatigue during repeated, high-intensity sprint exercise in competitive cyclists [26]
Beta-alanine	↑ cortisol level and ↓ testosterone:cortisol ratio at 24 h in a 24 h simulated military operation [40]	-better exercise performance and adaptation [41] ↑ muscle carnosine proportionally with the time of use [42,43] -no significant improvements in fatigue rates during high-intensity anaerobic exercise in highly trained athletes [44] -improved biochemical parameters in regards to muscle fatigue [45] ↑ exercise capacity [46] -elimination of executive function decline post-recovery [12] -improved performance, marksmanship and target engagement speed, as well as muscle endurance [47], nut no impact on cognitive performance [48]
Carbohydrates	-attenuate higher plasma cortisol levels and greater immune disturbances [49,50,51] -salivary cortisol has been observed to be similar in the self-selected diet group and carbohydrate group, but increased post-exercise in the self-selected diet group [52] ↓ salivary cortisol levels as a result of increased dietary carbohydrate as part of a Dietary Guideline for Americans-based diet [53] -a high-carbohydrate diet–higher salivary cortisol response in men [54]	
Sodium bicarbonate		-sodium bicarbonate in combination with caffeine led to a longer total distance of rowing compared to sodium bicarbonate alone or a possible additional benefit in 2000-m rowing performance when administered together with beta-alanine [55] -a small, but significant performance effect [56]
Probiotics		- strains have been reported to have an influence on exercise recovery [57,58] -beneficial effects in terms of intestinal permeability, immunity, microbiota, inflammation, GIs and URTIs [59] -*Lb. fermentum* in elite male distance runners ↓ risk and severity of URTIs [60], ↓ severity of GIT in cyclists [61] +*Lb. acidophilus*, *Lb. rhamnosus*, *Lb. casei*, *Lb. plantarum*, *B. lactis*, *B. breve*, *B.bifidum* and *Streptococcus thermophilus* ↑ run time to fatigue [62] *-B. breve* BR03 and *S. thermophilus* FP4 led to a positive effect on the reduced performance and range of motion followed by intense muscle damaging exercise [63] -an improvement in moderate-intensity exercise [64] *-Lb. plantarum* TWK10 reported significantly increased exercise performance in a dose-dependent manner [65] -*L. plantarum PS128* as a potential for better training management [66] *-B. longum* OLP-01 for performance-improving and health-promoting [67] -beneficial effects of probiotics in recovery with significantly increases at 24 and 72 h, as well as decreased soreness at 72 h post exercise [68]

## Data Availability

All the data is presented in the current article.

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
