# Peer review of "Connections between Different Sports and Ergogenic Aids—Focusing on Salivary Cortisol and Amylase"

_medicina, 2021, doi:10.3390/medicina57080753_

Round 1

Reviewer 1 Report

A more detailed explanation of the inclusion criteria, exclusion criteria, and selection process of journals included in the study is required.

Author Response

A more detailed explanation of the inclusion criteria, exclusion criteria, and selection process of journals included in the study is required.

Dear Reviewer 1,

Thank you for your suggestion to add more details regarding the inclusion and exclusion criteria for our study and we have added the information in the designated section (Lines 136-147).

Reviewer 2 Report

Overall, this paper needs to be reviewed by more English native speakers. However, this is a really good paper that can be salvaged since it is a timely review with a lot of good information.

Abstract: needs to be reviewed for proper English usage.

Introduction:

For line 44 please include the correlation coefficient for ease on the reader.

Due to the nature of this paper, this should be included in the introduction:

West, D. W., & Phillips, S. M. (2012). Associations of exercise-induced hormone profiles and gains in strength and hypertrophy in a large cohort after weight training. European Journal of Applied Physiology112(7), 2693-2702.

Additionally the introduction needs to be rewritten for language fluency.

Results:

This table has great information; however, it needs to be reformatted with the summary of each study condensed. Perhaps in a landscape rather than portrait style would be a good first step.

  1. Demographic groups and sports: for this section there needs to be written a more in-depth analysis of the outcomes of the studies. The point of this paper is to save time for the reader so that they do not need to investigate these papers further.
  2. Conclusions:

This sections needs to be more robust. Dissect the strengths and weaknesses of the nature of the studies covered. Perhaps have a limitations and future directions section.  What does this all mean for the practitioner?

Author Response

Dear Reviewer 2,

Thank you for your kind suggestions and we are pleased to say that we have considered and included them as follows:

Abstract: needs to be reviewed for proper English usage.

  • Thank you for your remark, we have reread the abstract and indeed there were significant changes needed to be done. We have modified it alongside other parts of the manuscript as well.
  •  

Introduction:

For line 44 please include the correlation coefficient for ease on the reader.

  • We are not entirely sure to which correlation coefficient you are referring to, however, we have added the one from the study cited at the line mentioned (it is a p<0.001) (Line 47)
  •  

Due to the nature of this paper, this should be included in the introduction:

West, D. W., & Phillips, S. M. (2012). Associations of exercise-induced hormone profiles and gains in strength and hypertrophy in a large cohort after weight training. European Journal of Applied Physiology112(7), 2693-2702

  • Thank for your insightful suggestion, the article is indeed very good and we did include it (discussing cortisol level from blood samples).

Additionally the introduction needs to be rewritten for language fluency.

  • Thank you for pinpointing that to us, we are happy to say that we also modified this section alongside the whole document. We hope that is more properly written this time around.

Results:

This table has great information; however, it needs to be reformatted with the summary of each study condensed. Perhaps in a landscape rather than portrait style would be a good first step

  • The table has been reformatted and switched to a landscape style, therefore it occupies less than 2 pages as of right now.

Demographic groups and sports: for this section there needs to be written a more in-depth analysis of the outcomes of the studies. The point of this paper is to save time for the reader so that they do not need to investigate these papers further.

  • Thank you for your kind suggestion. We modified the demographic subsection in order to add more information about the studies included so that the reader doesn’t need to go investigate the papers cited further (Lines 201-226).

Conclusions:

This sections needs to be more robust. Dissect the strengths and weaknesses of the nature of the studies covered. Perhaps have a limitations and future directions section.  What does this all mean for the practitioner?

  • Initially, we believed that the conclusion section does not need to be very robust as we have noted conclusions and observation at the end of each of the subsections from the results. However, after careful consideration, we have decided that it is indeed better to include these aspects in the conclusions, as well alongside limitations and future directions (Lines 451-489). Thank you for your kindness!

Round 2

Reviewer 2 Report

Thank you for the updates.